# Night Shift Work—A Risk Factor for Breast Cancer

**DOI:** 10.3390/ijerph17020659

**Published:** 2020-01-20

**Authors:** Marta Szkiela, Ewa Kusideł, Teresa Makowiec-Dąbrowska, Dorota Kaleta

**Affiliations:** 1Department of Hygiene and Epidemiology, Department of Hygiene and Health Promotion; Faculty of Health Sciences, Medical University of Lodz, 90-647 Łódź, Poland; marta.szkiela@umed.lodz.pl (M.S.); dorota.kaleta@umed.lodz.pl (D.K.); 2Department of Spatial Econometrics, Faculty of Economics and Sociology, University of Lodz, 90-255 Łódź, Poland; 3Department of Work Physiology and Ergonomics, Nofer Institute of Occupational Medicine in Łódź, 91-348 Łódź, Poland; tmd@imp.lodz.pl

**Keywords:** shift work, breast cancer, occupational choice, public health, value of life

## Abstract

Shift work is considered a risk factor for some health problems. This study aimed to determine whether night shift work is a significant factor for breast cancer risk. The case–control study was conducted from 2015–2019 in the Łódź region. The case group included 494 women diagnosed with malignant breast cancer; the control group included 515 healthy women. The research tool was an anonymous questionnaire. In the case group, the percentage of women working shift work was 51.9%, whereas in the control group, it was 34.1% (OR = 2.08 (95% CI: 1.61; 2.69)). A more insightful examination of shift work showed that only night work has a significant impact on breast cancer (OR = 2.61 (95% CI: 1.94; 3.53)). Even after considering the effect of other possible cancer factors (like high BMI, smoking, early menstruation, late menopause, pregnancy history, age, place of living, education), the odds of developing cancer were twice as high in the group of shift workers (OR = 2.20 (95% CI: 1.57; 3.08)). Considering a significantly higher risk of cancer among people working at night and a high percentage of such employees in Poland, the government should consider special action in the prophylactic treatment of cancers.

## 1. Introduction

In the European Union (EU) in 2018, 18.1% of employees worked shifts—slightly fewer than at the beginning of the period for which data on this topic are available in Eurostat (2002). The percentage of shift workers throughout the EU is much lower than in Poland, which in 2018 recorded 31.2% shift workers (Table 1). What is more, in the EU in 2018, 13.2% of the total working population worked night shifts, considerably fewer than in Poland, which in the same year recorded a figure of 17.5%. The percentage of working women is lower than men, which is understandable if we consider that shift work is primarily a characteristic of the industrial (in particular the mining section), transport, healthcare, and public administration sectors. Only in the healthcare sector do female shift workers dominate; all other sectors are mainly male. Although night work in the healthcare sector mainly concerns women (nurses), it is not very significant at the scale of the entire economy, because the whole healthcare sector (Section Q according to NACE rev. 2 classification) employs less than 6% of the total working population in Poland (LFS data for 2018), whereas the industrial sector (Sections B–E) employs 24.1%. 

The percentage of people working shifts (including at night) is sharply differentiated between EU countries (Figure 1). In 2018, 7.7% of the total working population in Lithuania worked night shifts, and in Slovakia this figure was almost three times higher (21.4%). Among all 28 EU countries, the highest percentages of night shift workers are found in Slovakia, Slovenia, Croatia, Malta, and Poland. Poland, which was ranked fifth among the 28 EU countries in 2018 in terms of percentage of people working night shifts, has previously ranked as high as second (2006–2007).

Working at night and exposure to light at night causes a disturbance of circadian rhythms. Among people working night shifts compared to those working only during the day, the following are more common: cardiovascular diseases, gastrointestinal disorders, peptic ulcers, diabetes, obesity, chronic fatigue and anxiety, sleep disorders, depression, and decreased melatonin secretion (through exposure to light at night) [1,2,3]. Among women, the incidence of menstrual disorders, reduced fertility, and increased risk of miscarriages, premature births, and low birth weight have been identified [4]. In addition, an increased risk of breast [2], prostate [5], endometrium [6], and large intestine [7] cancers, as well as non-Hodgkin’s lymphoma are indicated by existing studies [8]. 

Working at night and exposure to light at night causes a disturbance of circadian rhythms. Among the effects of circadian rhythm disturbances within the endocrine system, a change in the rhythms of secretion of many hormones, including prolactin, glucocorticoids, adrenocorticotropic hormone, corticoliberin, serotonin, and melatonin has been observed. Melatonin (N-acetyl-5-methoxytryptamine) is a hormone produced mainly by the pineal gland, and its secretion is dependent on light. Exposure to light inhibits melatonin secretion. Changes in melatonin synthesis and secretion may affect, among other things, the function of the immune system and the pituitary–thyroid and pituitary–gonadal axes as well as the hypothalamic–pituitary–adrenal axis [9,10]. The biosynthesis of melatonin also depends on the presence of tryptophan in the diet: an essential amino acid that is converted to serotonin and then metabolized to melatonin by the hydroxyindole-O-methyltransferase enzyme (HIOMT). Intestinal melatonin synthesis is not controlled by the biological clock, and it is stimulated by the presence of food [11]. Melatonin biosynthesis proceeds in a circadian rhythm depending on lighting conditions. Melatonin production is always highest at night. At night, the plasma melatonin concentration reaches its highest level (about 125 pg/mL), and during the day, it decreases significantly (to about 10 pg/mL) [12]. Melatonin secretion is induced by binding to nuclear receptors belonging to the RZR/ROR family and membrane receptors MT1, MT2, and MT3, the mRNA expressions of which fluctuate based on circadian rhythm, plasma melatonin concentration, and light intensity [13,14].

A potential relationship between insufficient melanin synthesis by the pineal gland and an increased risk of breast cancer was suggested for the first time by a group of scientists from the National Cancer Institute in the USA [15]. The first hypothesis regarding the relationship between breast cancer and night work and exposure to light at night was formulated in 1987 by Stevens. According to the Stevens hypothesis, work at night and exposure to light at night cause inhibition of melatonin synthesis, followed by an increase in estrogen levels, which can lead to an increased risk of developing breast cancer [16].

In 2007, the International Agency for Research on Cancer (IARC) classified night shift work as probably carcinogenic for humans (group 2A), based on sufficient evidence from animal studies and limited evidence from epidemiological studies [17]. The results of studies conducted since the publication of the IARC opinion remain ambiguous [18]. Hansen, in his review of the 26 existing studies, noted that there is some evidence that high numbers of consecutive night shifts have an impact on the extent of circadian disruption, and thereby there is an increased breast cancer risk and after over 20 years of night shift or after shorter periods with many consecutive shifts [19].

The aim of this study was to determine whether night shift work is a significant factor that increases the risk of cancer.

## 2. Material and Methods

### 2.1. Design and Study Population

The case–control study was conducted from 2015 to 2019 in the Łódź region. The case group included 494 women over 35 years old diagnosed with malignant breast cancer, after tumor resection or entire breast removal. The women were patients of the Oncological Surgery Department and the Second Department of Oncological Surgery, Oncological Surgery Clinics of the Provincial Specialist Hospital M. Kopernik in Łódź; the surgery department of Poddębice Health Center SP. Z O. O.; and the Provincial Specialist Hospital M. Skłodowska-Curie in Zgierz. The control group consisted of 515 healthy women without malignant breast cancer, similar in characteristics to the women from the study group in terms of age (±3 years) and education. The women from the control group were patients of the Provincial Specialist Hospital Maria Skłodowska-Curie in Zgierz, as well as clients of the club FruFitness Zgierz and the Adrianna stable in Aleksandrów Łódzki.

The criterion for including women in the study group was the presence of histopathologically confirmed malignant breast cancer (ICD-10: C.50.1–C.50.9) and no history of other cancers. The criterion for including women in the control group was no history of any cancers.

The research tool was an anonymous and voluntary questionnaire. The questions in the questionnaire concerned the following issues: sociodemographic data, pregnancy history, menstruation and menopause, type of contraception, hormonal treatment, physical activity, employment history, smoking and environmental smoke exposure, alcohol consumption, nutrition, family incidence of cancer, history of diseases, prenatal factors, and anthropometric data.

Before starting the main study, a pilot study was conducted among 15 patients from the case group and 15 women from the control group to check the reliability of the questionnaire.

The study design received a positive opinion from the Bioethics Committee at the Medical University of Łódź (RNN/236/15/EC of 22 September 2015).

### 2.2. Measures

#### 2.2.1. Outcome Variables

The outcome variable was breast cancer. The criterion for including women in the case group was the presence of histopathologically confirmed malignant breast cancer (ICD-10: C.50.1–C.50.9), with no history of other cancers. The criterion for including women in the control group was no history of any cancers.

Because the crucial predictor was shift work, the final number of respondents was limited to 973 women who answered questions about shift work (478 cases of breast cancer and 495 from the control group).

#### 2.2.2. Independent Variables

##### Sociodemographic Data

Respondent reported their birth date, age (which was analyzed in five variables: ≤47, 48–58, 59–69, 70–80, >80), place of living (countryside, small town (<50,000 inhabitants), medium town (51,000–100,000 inhabitants), large town (>100,000 inhabitants)), marital status (married, widow, never married, divorced, in separation) and education level (ISCED 5–6—higher education, ISCED 4—post-secondary education, ISCED 3+ secondary education, ISCED 3—vocational education, ISCED 1—primary education).

##### Anthropometric Data

Respondents reported their weight (kg) and height (m). Based on that information, BMI was calculated and categorized similarly to the WHO standard (Geneva, Switzerland). BMI was analyzed in four variables (<18, 18–25, 25–30, >30 kg/m^2^).

##### Pregnancy History

Respondents responded to a question on number of pregnancies. Number of pregnancies was analyzed in four variables (1, 2, 3, ≥4). Women also reported their age at their first delivery (<20, 20–27, ≥28) and duration (months) of breastfeeding (0, >6, 6–12, ≥12).

##### Menstruation and Menopause

Respondents reported their age of first menstrual period. Age of first menstrual period was analyzed in three variables (10–12, 13–15, 16–18). Women also reported their age of menopause (<40, 40–44, 45–49, 50–54, ≥55).

##### Smoking

Respondents were asked: “Have you ever smoked cigarettes?” If they replied that they had ever smoked, they were asked about their smoking status, with possible variants of responses “former smoker” or “active smoker”. Respondents were also asked about exposure to secondhand smoke.

##### Shift Work

Respondents were asked if they ever worked shifts, with possible answer options “Yes” and “No”. Those who reported working shifts were asked: "How many shifts did you work?" with possible answer options, “II” or “III”, where III meant night work. Of the 253 women who declared night shift work, the vast majority (88.5%) worked in this way for ten or more years before falling ill, and for not fewer than ten night shifts in the month (85.4%).

### 2.3. Statistical Analysis

All respondents’ answers were entered into a Microsoft Access database and then merged, ordered, and coded in a Microsoft Excel spreadsheet.

The following statistical methods were used in the analysis: (1) test z of proportion—to determine which categories significantly differentiated the control and study groups; and (2) the logit model to estimate the odds ratio of shift (night) work all other measured predictors controlled. Statistical calculations were carried out in (1) Microsoft Excel spreadsheet (frequency distribution of the variables, contingency tables, OR for 2 × 2 tables); (2) Gretl econometric package (test of proportion and logit model selection); and (3) Statistica 13.3 (OR and CI for the final version of the logit model).

Predictor reduction in the logit model was carried out using a backward elimination model selection procedure, which started with the most general model and eliminated one variable at a time until the best model was reached (i.e., when all right-side variables were statistically significant for *p* ≤ 0.05).

## 3. Results

Considering the aim of the study (to determine whether night shift work is a significant factor in the development of breast cancer), the analysis covered 973 women who answered the question regarding shift work (out of 1009). Among these women, there were 495 (50.8%) patients with breast cancer and 478 (49.2%) healthy women (without breast cancer). Of the 253 women who declared night shift work, the vast majority (88.5%) had worked in this way for ten or more years before falling ill and for not fewer than ten night shifts in the month (85.4%).

The characteristics of the examined women are presented in Table 2, in which the test of proportion was used to determine the significance of differences between the control group (women not suffering from breast cancer) and the case group (women who had breast cancer).

Categories that significantly differentiated groups (case and control) are marked in bold. As is evident, the case and control groups differed in nearly all variables (the exception being the age at the first childbirth). Most of these differences were due to well-documented factors of breast cancer: the risk of developing the disease increases with age, high BMI, young age of the first menstruation, late age of menopause, multiplicity of pregnancies, short period of breastfeeding (or lack of breastfeeding). In addition, social factors significantly differentiated the groups: a higher percentage of the patient group was made up of people living in the village, widows, and people with low education (ISCED 3−, ISCED 1).

As indicated in Table 2, the fact of night shift work significantly differentiated the control and case groups (*p* < 0.001). In the case group, the percentage of women working shifts was 51.9%, whereas it was 34.1% in the control group. The odds ratio was OR = 2.08 (95% CI: 1.61; 2.69), meaning that the odds of developing cancer was twice as high (precisely 108% higher) in the group of shift workers than those not working shifts.

A more insightful examination of shift work (or, more precisely, the separation of shift workers into people working in a two-shift system and a three-shift system) showed that only night work had a significant impact on the risk of breast cancer. For two-shift work, there were identical percentages of employees in both groups, indicating that this type of work was not significantly different between groups. However, for night work, an OR of 2.61 (95% CI: 1.94; 3.53) was observed.

Finally, a logit model was built where, apart from night work, all other factors included in Table 2 which significantly differentiated the case and control groups were considered. The backward elimination procedure was used to reduce the model to a form with only statistically significant predictors (*p* ≤ 0.05 significance level). A list of these variables, ORs, and 95% confidence intervals is shown in Table 3.

Table 3 shows that night shift work more than doubles the odds of breast cancer (OR = 2.2; 95% CI 1.57–3.08), but the most significant factor for cancer was BMI. In addition, the factors of not breastfeeding or short breastfeeding time, early menstruation, and late menopause more than doubled the odds of getting sick. Two sociodemographic factors—living in the countryside and widow status—turned out to be less strong than previously suggested, but remained significant factors in the development of breast cancer. The factor that reduced the odds the most (by 60%) was not smoking.

## 4. Discussion

The results of published epidemiological studies on breast cancer in relation to shift work have been inconsistent. Some studies have confirmed an increased risk of breast cancer in shift workers, while others have not [20,21,22,23,24,25,26]. Cohort studies have been mainly limited to one professional group with a specific work time pattern (e.g., nurses), and have less frequently included representatives of other professions, mainly from industry (e.g., textile industry). Our study did not limit to one professional group. Meta-analyses of current epidemiological studies have reported increases of approximately 20% in summed relative risks of breast cancer in women who work/worked shifts, compared to women working only during the day [27,28,29]. One of the first studies confirming the hypothesis that shift work might increase the risk of breast cancer was a case–control study conducted in Denmark, which found that the relative risk of breast cancer in shift workers was 1.5 and in shift workers of over 6 years it was 1.7, demonstrating the direct relationship between shift work and an increased risk of breast cancer [30]. In the United States, the Nurses’ Health Study (NHS) prospective cohort study covered 78,586 nurses working shifts (at least three night shifts a month). During the 10 years of observation, 2441 cases of breast cancer were registered. It was found that the risk of breast cancer increased from approximately 8% in women doing shift work (including working at night) from 1 to 29 years, up to 36% in women doing shift work for over 30 years [31]. In the subsequent edition of Nurses’ Health Study II (NHS II), the studied cohort consisted of 115,022 pre-menopausal nurses. During a 12 year follow-up (1989–2001), 1352 cases of invasive breast cancer were diagnosed. Researchers observed a 79% increased risk of developing breast cancer in nurses with at least 20 years of work experience in a rotational shift system (at least three night shifts a month) compared to nurses who had never worked shift work [32]. The hypothetical relationship between breast cancer and circadian rhythm disturbances was also assessed in the Electromagnetic Fields and Breast Cancer Long Island case–control study. The control group consisted of 576 women with breast cancer diagnosed in the period between August 1996 and June 1997, whereas the control group was 585 healthy women. Both groups were <75 years old and had lived in Long Island, New York ≥15 years. The research tool was a personal questionnaire assessing light exposure at night during shift work (for 15 years) and at home (for 5 years). There was no statistically significant relationship between shift work and an increased risk of breast cancer. Instead, an increased risk was found for women exposed to light at night at home (OR = 1.65, 95% CI: 1.02–2.69) [33]. In the analysis of the case–control study (CECILE), Menegaux et al. found that working in the night shift system increased the risk of developing breast cancer. It was observed that 311 women (12%) had worked shifts. The odds of developing breast cancer associated with night work (OR = 1.35 (CI = 1.01–1.80)) was higher than that associated with work in the late evening (OR = 1.25 (CI = 0.79–1.98)). The morning change was not associated with a risk of developing breast cancer. Night work lasting 4.5 years or more was associated with an odds ratio (OR) of 1.40 (CI = 1.01–1.92). The OR for women working at night, on average fewer than three nights a week, was 1.43 (CI = 1.01–2.03), whereas for women who worked at night more than three nights a week, the OR was 1.14 (CI = 0.82–1.59). In the analyses combining night work and the average number of nights worked per week, the relationship with the risk of developing breast cancer was particularly evident among women who had been working in a night shift system ≥4.5 years, working on average fewer than three nights a week (OR = 1.83 (CI = 1.15–2.93)). The risk of developing breast cancer in women who had ever worked the night before the first delivery of a live baby was higher (OR = 1.47 (CI = 1.02–2.12)) compared to women who had never worked in the night. The OR for women who started work at night after the first delivery with a live baby was 1.09 (CI = 0.77–1.55). The OR for night work before the first birth giving birth to a live child was greater among women who worked at night for more than 4 years before the first birth giving birth to a live child (OR = 1.95 (CI = 1.13–3.35)), and those who worked at night, during this period of life, for less than three nights a week on average (OR = 2.24) CI = 1.15–3.71)). After combining the duration and frequency of night work before the first delivery ending with the birth of a live child, it was found that for night work lasting over four years and fewer three nights a week, the OR was 3.03 (CI = 1.41–6.50) [20]. Fritschi et al. conducted a case–control study in France in which they observed evidence that some factors associated with night shift work may be associated with an increased risk of breast cancer. A slight increase in the risk of developing breast cancer was observed in women working at night (work from midnight to 5 a.m.) (OR = 1.16, 95% CI = 0.97–1.39). In the case of shift workers, there was a 22% increase in the risk of developing breast cancer (OR = 1.22, 95% CI = 1.01–1.47) with a statistically significant dose–response relationship (*p* = 0.04) [4]. Based on the analysis of data obtained on the basis of the German case–control study (GENICA) carried out by Pesch et al., it was found that long-term work in the night shift system is associated with a small increase in the risk of developing breast cancer. It was observed that approximately 13% of all women were ever employed in shifts. Almost the same number of women from the study (6.5%) and control (6.4%) groups had performed night work for more than or equal to one year. Among the women from the counter group working at night shifts, 63.0% worked in healthcare. Women working at night were younger (average age 54), especially when compared to women who had never worked at night (average age 68). Compared to women working only during the day, women working at night were more childless (28.6% vs. 17.8%), had lower levels of education (12.3% vs. 9.2%), and less frequently used hormone therapy (35.7% vs. 51.9%). It was found that the increased risk of developing breast cancer was not associated with shift work or night work ever, compared with women working only during the day (OR = 0.96; 95% CI = 0.67–1.38 and OR = 0.91, 95% CI = 0.55–1.49). For women who worked for more than 807 nights, the odds of developing breast cancer was significantly higher (OR = 1.73 95% CI 0.71–4.22). Night work for 20 years or more was associated with an OR of 2.48 (95% CI 0.62–9.99) [23].

The studies listed above showed different results regarding the impact of shift work on breast cancer. Unfortunately, they are not comparable due to the different sets of variables, definitions of these variables, statistical methods used, sample size etc. Our study seems to be the first Polish attempt to estimate the exact impact of shift work on the odds of cancer (by calculating OR coefficient in the multifactor logit regression). Research on the health consequences of shift work is sometimes inconclusive. In this context, our study is a voice in this discussion, supporting the hypothesis of the harmful effects of night shift work, which can increase the risk of developing breast cancer. The present study showed that night shift work more than doubles the odds of breast cancer (OR = 2.2; 95% CI 1.57–3.08).

The strengths of the study were a large study group, many analyzed variables and an extensive questionnaire. The weaknesses of the study include the fact that the questionnaire questions concerned distant events in the past of the surveyed women. The respondents did not always accurately remember events from 10–15 years ago.

## 5. Conclusions

Our study supported the hypothesis that night shift work is a significant risk factor for breast cancer. It is worth emphasizing that only night shift work increases this risk (shift work, if it is not at night—does not affect the risk). Eurostat statistics show that 13.2% of Europeans and 17.5% of Poles work night shifts, so the population exposed to increased risk of cancer is substantial, especially in Poland. Considering these two facts—a significantly higher risk of cancer among people working at night and the high percentage of such employees in Poland—the government should consider the special action in the prophylactic treatment of cancers. It is necessary to consider introducing changes to the shift work system in Poland, especially among women. Limit the possibility of night work among women over the age of 45, who are at the highest risk of developing breast cancer, should be considered. Particular attention should be paid to compliance with the principles of health and safety at work, especially the appropriate number of breaks and ergonomic working conditions.

## Figures and Tables

**Figure 1 ijerph-17-00659-f001:**
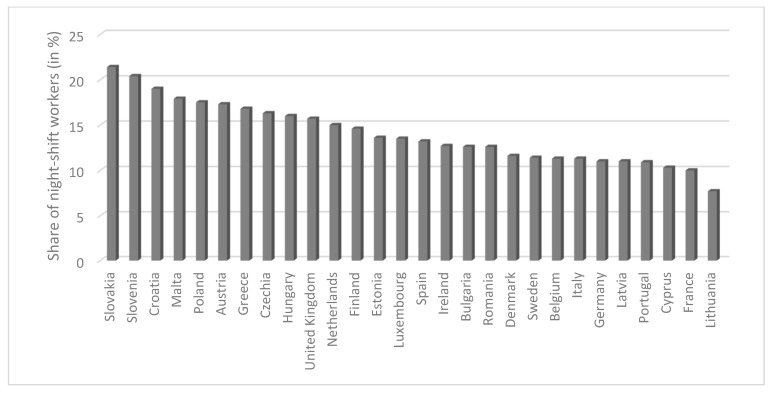
Employed persons working at night as a percentage of the total employment in each EU28 country in 2018 (based on Eurostat). Source: own elaboration based on Eurostat data (tables titled *lfsa_ewpnig*).

**Table 1 ijerph-17-00659-t001:** Employees working shifts and nights as a percentage of the total of employees (based on Eurostat).

Year	Shift Work				Night Work			
	EU28		Poland		EU28		Poland	
	Total %	Female %	Total %	Female %	Total %	Female %	Total %	Female %
2001			38.8	33.4			21.3	13.6
2002	18.9	16.6	36.8	32.0			20.7	12.9
2003	18.7	16.5	36.9	32.6			20.9	13.2
2004	17.9	16.1	34.3	30.3	16.1	10.8	20.9	12.8
2005	18.1	16.2	35.9	32.0	16.3	10.7	21.6	13.5
2006	17.4	15.7	30.8	27.3	15.8	10.6	21.8	13.8
2007	17.9	16.5	29.1	26.8	15.7	10.8	21.3	14.2
2008	17.3	16.2	29.3	28.1	14.8	10.1	19.9	13.6
2009	16.9	15.6	29.4	28.2	14.8	10.2	18.2	12.4
2010	17.4	16.1	29.9	28.5	14.9	10.3	17.9	11.9
2011	17.6	16.2	29.9	28.5	14.9	10.3	17.7	11.9
2012	17.7	16.3	30.3	29.2	14.4	9.9	18.3	12.4
2013	17.8	16.4	30.5	29.2	13.8	9.7	18.9	12.8
2014	18.1	16.7	30.8	29.6	13.8	9.6	18.3	12.3
2015	18.3	17.0	30.8	30.0	13.8	9.8	17.6	12.4
2016	18.5	17.2	31.7	30.9	13.8	9.8	18.0	12.6
2017	18.2	17.0	31.5	30.4	13.3	9.5	17.4	12.2
2018	18.1	16.8	31.2	29.9	13.2	9.4	17.5	12.4

Source: own elaboration based on Eurostat data (tables titled *lfsa_ewpshi* and *lfsa_ewpnig*).

**Table 2 ijerph-17-00659-t002:** Characteristics of the cases and control groups *.

Categories	Control	Case	Control	Case	*p*-Value
	N	N	%	%	
	Age				
47−	52	46	10.5%	9.6%	0.648
48–58	151	117	30.5%	24.5%	0.036
59–69	202	193	40.8%	40.4%	0.891
70–80	73	95	14.7%	19.9%	0.035
>80	17	27	3.4%	5.6%	0.097
sum	495	478	100%	100%	
	Place of living				
Countryside	111	150	23.9%	32.3%	0.005
Small town (<50 thou.)	88	74	18.9%	15.9%	0.226
Medium town (51–100 thou.)	167	128	35.9%	27.5%	0.006
Large town (>100 thou.)	99	113	21.3%	24.3%	0.274
sum	465	465	100.0%	100.0%	
	Marital status				
Married	367	326	76.3%	71.0%	0.067
Widow	45	75	9.4%	16.3%	0.001
Never married	41	31	8.5%	6.8%	0.308
Divorced	25	24	5.2%	5.2%	0.983
In separation	3	3	0.6%	0.7%	0.954 *
sum	481	459	100.0%	100.0%	
	Education				
ISCED 5_6	150	94	30.5%	19.7%	0.000
ISCED 4	22	24	4.5%	5.0%	0.682
ISCED 3+	190	171	38.6%	35.8%	0.373
ISCED 3−	85	118	17.3%	24.7%	0.004
ISCED 1	45	70	9.1%	14.7%	0.008
sum	492	477	100.0%	100.0%	
	BMI				
<18	13	3	2.7%	0.6%	0.014 *
18–25	352	199	72.3%	41.9%	<0.001
25–30	84	170	17.2%	35.8%	<0.001
>30	38	103	7.8%	21.7%	<0.001
sum	487	475	100.0%	100.0%	
	Age of first menstrual period				
10–12	111	184	22.5%	38.9%	<0.001
13–15	302	239	61.3%	50.5%	0.001
16–18	80	50	16.2%	10.6%	0.010
sum	493	473	100.0%	100%	
	Age of menopause				
<40	4	1	1.1%	0.3%	0.149 *
40–44	18	25	5.2%	6.6%	0.416
45–49	28	45	8.0%	11.9%	0.087
50–54	230	177	66.1%	46.7%	<0.001
≥55	68	131	19.5%	34.6%	<0.001
sum	348	379	100.0%	100.0%	
	Number of pregnancies				
0	80	63	16.2%	13.2%	0.194
1	154	110	31.1%	23.1%	0.005
2	169	177	34.1%	37.1%	0.335
3	70	88	14.1%	18.4%	0.069
≥4	22	39	4.4%	8.2%	0.017
sum	495	477	100.0%	100.0%	
	Age of first delivery				
<20	64	65	15.6%	15.9%	0.900
20–27	288	293	70.1%	71.6%	0.622
≥28	59	51	14.4%	12.5%	0.428
sum	411	409	100.0%	100.0%	
	Duration (months) of breastfeeding				
0	95	157	21.5%	35.6%	<0.001
<6	111	156	25.1%	35.4%	0.001
6–12	221	119	50.0%	27.0%	<0.001
>12	15	9	3.4%	2.0%	0.217 *
	442	441	100.0%	100.0%	
	Smoking				
Former smoker	113	162	38.6%	43.1%	0.239
Non-smoker	43	17	14.7%	4.5%	<0.001
Smoker	58	82	19.8%	21.8%	0.526
Passive smoker	79	115	27.0%	30.6%	0.306
sum	293	376	100.0%	100.0%	
	Shift workers				
No	326	230	65.9%	48.1%	<0.001
Yes	169	248	34.1%	51.9%	<0.001
sum	495	478	100.0%	100.0%	
	II shift workers				
No	410	396	82.8%	82.8%	0.994
Yes	85	82	17.2%	17.2%	0.994
sum	495	478	100.0%	100.0%	
	III (night) shift workers				
No	410	310	82.8%	64.9%	<0.001
Yes	85	168	17.%	35.1%	<0.001
sum	495	478	100.0%	100.0%	

Source: own calculations in Microsoft Excel and Gretl. * indicates that not all assumptions of proportion test were met and the result should be treated with caution.

**Table 3 ijerph-17-00659-t003:** Statistically significant odds ratios (ORs) (ordered in descending order).

Variable	OR	95% CI	
BMI > 30	3.60	2.34	5.56
BMI = 25–30	3.07	2.18	4.32
breast-feeding = 0	2.96	2.07	4.23
breast-feeding < 6 months	2.87	2.00	4.11
menstruation in age: 10–12	2.35	1.70	3.25
night shift work	2.20	1.57	3.08
menopause age: 55+	2.06	1.42	2.98
widow	1.67	1.07	2.61
countryside	1.65	1.18	2.31
non-smoker	0.40	0.21	0.77

Source: own calculations in Gretl and Statistica.

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
