# Peer review of "Night Shift Work—A Risk Factor for Breast Cancer"

_ijerph, 2020, doi:10.3390/ijerph17020659_

Round 1

Reviewer 1 Report

This paper aims to study the associtions between shift and night work and risk of breast cancer in Poland.

In general the paper is somewhat superficial described and lack detailed definition of the study population and the exposures as well as statistical analysis.

Abstract:

Information on proportion of workers with shift/night work could be omitted

Introduction:

The paper should focus on breast cancer only. The IARC evaluation was updated June 2019 The most recent review of night work and breast cancer should be added: Hansen J. NightShift Work and Risk of Breast Cancer. Environ Health Rep. 2017 Sep;4(3):325-339.

Material and Methods:

It is not described how the control subjects were retrieved. From where were they recruited, age distribution, etc.? A description of response rates for cases and controls are needed. What is the definition of shift and night work? The question “How many shifts did you work?” is unclear The statistical analyses is unclear

Results:

The duration and type of night/shift work should be included in the analysis It is uncertain if the variables in Table 4 re mutually adjusted

Discussion:

The existing studies of breast cancer have different level of quality. Therefore, the results appears somewhat inconsistent. This should be highlighted. It should be discussed that the present study shows a higher relative risk than seen in other similar studies.

Author Response

Dear reviewer,
Thank you for all the comments in the text. We answer them below.

Abstract:

Information on proportion of workers with shift/night work could be omitted

Information on proportion of workers with shift/niight work has been omitted. 

Introduction:

The paper should focus on breast cancer only. The IARC evaluation was updated June 2019 The most recent review of night work and breast cancer should be added: Hansen J. NightShift Work and Risk of Breast Cancer. Environ Health Rep. 2017 Sep;4(3):325-339.

New information on breast cancer has been added to the introduction as well as Hansen paper.

Material and Methods:

It is not described how the control subjects were retrieved. From where were they recruited, age distribution, etc.? A description of response rates for cases and controls are needed. What is the definition of shift and night work? The question “How many shifts did you work?” is unclear The statistical analyses is unclear

We added the description of how the control subjects were retrieved as well as a more precise definition of night shift. Age and other socio-demographic distribution is given in table 3.

Results:

The duration and type of night/shift work should be included in the analysis It is uncertain if the variables in Table 4 re mutually adjusted.

The information that "the vast majority (88.5%) worked in this way for ten or more years before falling ill and for not less than ten night shifts in the month (85.4%)" has been added to "shift work" section.

Discussion:

The existing studies of breast cancer have different level of quality. Therefore, the results appears somewhat inconsistent. This should be highlighted. It should be discussed that the present study shows a higher relative risk than seen in other similar studies.

The proper paragraph has been added in discussion.

Reviewer 2 Report

This is another study that supports the hypothesis that night shift work is associated to breast cancer. Recent studies (i.e. Jones, BJC 2019) and a meta analysis (Cordina-Duverger  Eur J Epidemiol 2018) show inconsistent results.  These discrepancies are basically due to differences in methods and more specifically in exposure assessment.  

This case control study was well conducted but more in deep description and analysis of how night work exposure was investigated is required in order to provide more evidence that there was not misclassification bias.

Authors mention that they evaluated exposure wuth the following question:

Respondents were asked if they ever worked shift, with possible answer options: „Yes”, „No”. Those who reported working shifts were asked "How many shifts did you work?" with possible answer options: "II", "III".

How was questioned the duration and timing of night shift work?

Author Response

Dear reviewer,
Thank you for all the comments in the text. We answer them below.

This is another study that supports the hypothesis that night shift work is associated to breast cancer. Recent studies (i.e. Jones, BJC 2019) and a meta analysis (Cordina-Duverger  Eur J Epidemiol 2018) show inconsistent results.  These discrepancies are basically due to differences in methods and more specifically in exposure assessment.  

This case control study was well conducted but more in deep description and analysis of how night work exposure was investigated is required in order to provide more evidence that there was not misclassification bias.

Thank you for these remarks. We added new information in the introduction and discussion section.

Authors mention that they evaluated exposure wuth the following question:

Respondents were asked if they ever worked shift, with possible answer options: „Yes”, „No”. Those who reported working shifts were asked "How many shifts did you work?" with possible answer options: "II", "III".

How was questioned the duration and timing of night shift work?

Answer III meant night work. Of the 253 women who declared night shift work, the vast majority (88.5%) worked in this way for ten or more years before falling ill and for not less than ten night shifts in the month (85.4%).

This information has been added to the description of shift work.